# Deep Characterization of Circular RNAs from Human Cardiovascular Cell Models and Cardiac Tissue

**DOI:** 10.3390/cells9071616

**Published:** 2020-07-04

**Authors:** Tobias Jakobi, Dominik Siede, Jessica Eschenbach, Andreas W. Heumüller, Martin Busch, Rouven Nietsch, Benjamin Meder, Patrick Most, Stefanie Dimmeler, Johannes Backs, Hugo A. Katus, Christoph Dieterich

**Affiliations:** 1Section of Bioinformatics and Systems Cardiology, Klaus Tschira Institute for Integrative Computational Cardiology, University Hospital Heidelberg, 69120 Heidelberg, Germany; Jessica.Eschenbach@med.uni-heidelberg.de; 2Department of Internal Medicine III (Cardiology, Angiology, and Pneumology), University Hospital Heidelberg, 69120 Heidelberg, Germany; Martin.Busch@med.uni-heidelberg.de (M.B.); Rouven.Nietsch@med.uni-heidelberg.de (R.N.); Benjamin.Meder@med.uni-heidelberg.de (B.M.); Patrick.Most@med.uni-heidelberg.de (P.M.); Hugo.Katus@med.uni-heidelberg.de (H.A.K.); 3German Centre for Cardiovascular Research (DZHK)-Partner Site Heidelberg/Mannheim, 69120 Heidelberg, Germany; Johannes.Backs@med.uni-heidelberg.de; 4Institute of Experimental Cardiology, University Hospital Heidelberg, 69120 Heidelberg, Germany; Dominik.Siede@gmail.com; 5Institute for Cardiovascular Regeneration, Centre for Molecular Medicine, Goethe University Frankfurt, 60590 Frankfurt, Germany; Heumueller@med.uni-frankfurt.de (A.W.H.); Dimmeler@em.uni-frankfurt.de (S.D.); 6German Centre for Cardiovascular Research (DZHK)-Partner site Rhine/Main, 60590 Frankfurt, Germany

**Keywords:** circRNAs, hiPSC-CMs, HUVEC, AUG circRNAs, RNase R, conservation, m^6^A-methylation

## Abstract

For decades, cardiovascular disease (CVD) has been the leading cause of death throughout most developed countries. Several studies relate RNA splicing, and more recently also circular RNAs (circRNAs), to CVD. CircRNAs originate from linear transcripts and have been shown to exhibit tissue-specific expression profiles. Here, we present an in-depth analysis of sequence, structure, modification, and cardiac circRNA interactions. We used human induced pluripotent stem cell-derived cardiac myocytes (hiPSC-CMs), human healthy and diseased (ischemic cardiomyopathy, dilated cardiomyopathy) cardiac tissue, and human umbilical vein endothelial cells (HUVECs) to profile circRNAs. We identified shared circRNAs across all samples, as well as model-specific circRNA signatures. Based on these circRNAs, we identified 63 positionally conserved and expressed circRNAs in human, pig, and mouse hearts. Furthermore, we found that the sequence of circRNAs can deviate from the sequence derived from the genome sequence, an important factor in assessing potential functions. Integration of additional data yielded evidence for m^6^A-methylation of circRNAs, potentially linked to translation, as well as, circRNAs overlapping with potential Argonaute 2 binding sites, indicating potential association with the RISC complex. Moreover, we describe, for the first time in cardiac model systems, a sub class of circRNAs containing the start codon of their primary transcript (AUG circRNAs) and observe an enrichment for m^6^A-methylation for AUG circRNAs.

## 1. Introduction

Despite significant advances in treatments, cardiovascular disease (CVD) is still the leading cause of death in developed countries [1]. With the broad applicability of omics methods, genetic components have been identified in CVD subtypes such as cardiomyopathy [2] and coronary artery disease [3]. Furthermore, transcriptional and post-transcriptional regulation, such as alternative mRNA splicing, have been implicated as cause, as well as, the consequence of CVD [4]. Transcript back-splicing leads to circular RNA (circRNA) formation and has recently gained attention in different organisms and tissues [5,6,7].

circRNAs are covalently closed rings, which typically are not polyadenylated and therefore are resistant to exonucleases. Furthermore, the expression of circRNAs is highly specific for cell type and developmental stage, while additionally being independent from the expression of the linear host gene in many cases [8].

Due to their stability, circRNAs are seen as potential biomarkers, especially in cancer [9], but also for heart failure [10], since circRNAs have also been detected in the human blood stream [11].

From a mechanistic view, we are still only beginning to understand the functions of circRNAs. For a few selected circRNAs the ability to bind miRNAs on multiple binding sites was shown [12,13]. In a similar way, circRNAs also have been shown to be involved in a range of interactions with RNA binding proteins (RBPs) [14]. More recently, first reports of translated circRNAs have been published [15,16], thus adding another facet to this versatile RNA class. However, these detailed studies only cover a small fraction of the large, known repertoire of circRNAs, leaving the vast majority of circRNAs without assigned function.

Detection of circRNAs is primarily facilitated by locating sequencing reads that span the back-splice junction (BSJ) between the two joined exons. This fraction of reads is the only class that can clearly be assigned to a circRNA, while all other reads that map to a potential exon of the circRNA cannot be specifically assigned to a circRNA. Thus, most circRNA detection algorithms focus on retrieving BSJ-spanning reads (see [17,18] for circRNA detection reviews).

While computational detection of circRNAs is generally feasible with sequence data originating from total RNA libraries after depletion of ribosomal RNA, optimized approaches like CircleSeq have been developed [19]. Here, the RNA pool is divided: one portion is treated with RNase R, thus digesting linear RNA fragments, while the other portion receives a mock treatment (Figure 1A).

We [20] and others [21,22] provided a first comprehensive catalog of circRNAs in rat, mouse, and human, including first studies of developmental expression of circRNAs in human induced pluripotent stem cell-derived cardiac myocytes (hiPSC-CMs) [8,23].

In this study, we compare and complement deep sequencing data from hiPSC-CMs with data from human umbilical vein endothelial cells (HUVECs). By including data from HUVECs we furthermore intend to incorporate aspects of the endothelium in the progression of CVD into our study [24]. Several CVD risk factors, such as hypertension, atherosclerosis, and diabetes mellitus can be studied in HUVECs [25]. Thus, circRNAs expressed in HUVECs are a plausible extension to hiPSC-CM-expressed circRNAs when CVDs are in focus. These two cardiovascular cell models are accompanied by deep sequencing data from healthy and diseased human cardiac tissue to form a comprehensive core dataset, covering human cell models of CVD as well as patient tissue from healthy and diseased states.

The overarching goal of this study is to reveal the exon composition of circRNAs in the aforementioned systems. Here, for the first time, we resolve the exon composition of cardiac circRNAs, which is also subject to alterations across cell types and tissues.

To assess possible functions of circRNAs, we included additional data sets such as ribosomal footprinting data [26], m^6^A-methylation RNA immunoprecipitation data [27], and Argonaute 2 (Ago2) cross-linking immunoprecipitation (CLIP) data [28]. While the ribosomal footprinting data informs us about circRNAs that interact with ribosomes, Ago2 CLIP data helps to identify candidate circRNAs, which may interact with the RISC complex (miRNA-target complex).

Moreover, we aimed to provide researchers with validated lists of conserved circRNAs. In this study, these conserved circRNAs were computationally mapped between genomes of routinely employed model species for in vivo cardiovascular research, such as mouse and pig. Furthermore, these computationally mapped circRNAs were cross-verified using CircleSeq data from pig and mouse to assure circRNAs mapped from one species to another are indeed expressed in both species.

## 2. Materials and Methods

### 2.1. hiPSCs Maintenance Culture

The hiPSC line was obtained from Bruce R. Conklin (Gladstone Institute, University of California), called here WTC11. The hiPSCs were cultured in Essential 8 Medium (Thermo Fisher Scientific, Bremen, Germany; Cat# A1517001) until reaching a confluence of around 70% on Matrigel Matrix-coated plates, (BD Bioscience, San Jose, CA, USA; Cat# 354277). Matrigel Matrix was diluted in RPMI1640 as indicated in manufacture’s protocol. For splitting, the hiPSCs were washed with PBS and incubated with Accutase for ∼10 min until detachment. Cells were collected in PBS with a volume of five times the used Accutase. Cells were centrifuged for 3 min at 300× *g*. Cells were resuspended in 5 mL Essential 8 Medium containing 10 μmol ROCK inhibitor (Tocris, Wiesbaden-Nordenstadt, Germany; Cat# 1254). The cells were counted in a Neubauer chamber and plated on Matrigel Matrix-coated plates at a density of 20,000 cells/cm${}^{2}$.

### 2.2. hiPSCs Differentiation towards Cardiac Myocytes (hiPSC-CMs)

The hiPSCs were differentiated as previously described [29,30]. The hiPSCs were harvested from Matrigel Matrix-coated 6-well plates at ∼90% confluence and seeded in 12-well plates coated with Matrigel Matrix in Essential 8 Medium and 10 μmol ROCK inhibitor. Matrigel Matrix was diluted in RPMI1640 as indicated in manufacturer’s protocol and Matrigel Matrix-coated plates were incubated at least for 1 h at 37 ${}^{\circ }$C and 5% CO${}_{2}$ in humidified atmosphere. The hiPSCs were cultured for 3 days in Essential 8 Medium until a confluence of 70–90%. Three days after plating, the medium was changed to RPMI1640 with B27 Supplement minus insulin (Thermo Fisher Scientific, Bremen, Germany; Cat# A1895601) and 10 μmol CHIR99021 (Tocris, Wiesbaden-Nordenstadt, Germany; Cat# 4953) for exactly 24 h. This is referred to as day 0. On day 1, the medium was changed to RPMI1640 with B27 Supplement minus insulin. On day 3, 5 μmol inhibitor of WNT production 2 (IWP2) in RPMI1640 with addition of B27 Supplement minus insulin was added to the cells. On day 5, media was changed to RPMI1640 plus B27 Supplement minus insulin. Beyond day 7, the medium was changed every other day to RPMI1640 with B27 Supplement until day 15. First beating was observed on day 8. On day 15, the hiPSC-CMs were trypsinized. The cells were collected by centrifugation at 300× *g* for 5 min. The cell pellet was resuspended in RPMI1640 plus B27 Supplement and 10 μmol ROCK inhibitor. The harvested cells from 12-well plate were plated in a Matrigel Matrix-coated T75 cm${}^{2}$ flask. After 3 days in flasks, the medium was changed to lactate medium to start the metabolic purification process. Cells were used for experiments when two subsequent treatments of lactate for 4 days were done. For quality control, flow cytometry analysis of cardiac TnnT2 was done after lactate purification showing a pure cardiac myocyte population of ∼70–95%.

### 2.3. HUVEC Cell Culture

Pooled human umbilical vein endothelial cells (HUVECs) were purchased from Lonza and cultured in endothelial basal media (Lonza, Basel, Switzerland; Cat# CC-3121) supplemented with EGM Single Quots (Lonza, Basel, Switzerland; Cat# CC-4133) and 10% fetal bovine serum (Thermo Fisher Scientific, Bremen, Germany; Cat #10270-106). Cells were kept in a humidified incubator at 37 ${}^{\circ }$C and 5% CO${}_{2}$. HUVECs were cultured in cell culture flasks (Greiner, Kremsmünster, Austria) until the third passage and harvested afterwards.

### 2.4. Pig Tissue Samples

Porcine left ventricular samples were isolated from 10–12-week-old German farm pigs (Braeunling; body weight 28–37 kg) that underwent post-ischemic heart failure induction by temporary (2 h) occlusion of the proximal ramus circumflexus, as previously described in [31]. Animals were sacrificed 2 weeks post-infarction and transmural left ventricular samples for subsequent RNA isolation were extracted from infarcted area, as well as remote myocardium. All animal procedures and experiments were performed in accordance with the German regulations for animal welfare and were approved by the local Animal Care and Use Committee of Baden-Wuerttemberg.

### 2.5. RNA Extraction from Human Hearts

Biopsy specimens were obtained from the apical part of the free left ventricular wall from patients undergoing cardiac catheterization using a standardized protocol. Biopsies were rinsed with NaCl (0.9%) and immediately transferred and stored in liquid nitrogen until RNA was extracted. Total RNA was extracted from biopsies using the Allprep DNA/RNA kit (Qiagen, Hilden, Germany; Cat# 80204) according to the manufacturer’s protocol. RNA purity and concentration were determined using the Fragment Analyzer (Agilent Technologies, Waldbronn, Germany) with a DNF-471 standard sensitivity RNA assay.

### 2.6. RNA Extraction from HUVECs

For RNA extraction, HUVECs were washed once with PBS and cells were scraped and harvested in Qiazol (Qiagen, Hilden, Germany; Cat# 1038703). RNA was then isolated by phenol-chloroform extraction using the miRNeasy Kit (Qiagen, Hilden, Germany; Cat# 1038703) according to the manufacturer’s instructions and including an additional DNase I (Qiagen, Hilden, Germany; Cat# 79254) digestion step. RNA was eluted in water and stored at −80 ${}^{\circ }$C.

### 2.7. RNA Extraction from hiPSC-CMs

RNA extraction was done from hiPSC-CMs with TRI Reagent (Sigma-Aldrich, Darmstadt, Germany; Cat# T9424), as described in the manufacturer’s protocol. Briefly, 1 mL TRI Reagent was added per well to the attached cells and incubated for 5 min at room temperature. To form a homogeneous lysate, the cells were pipetted several times with a P1000 pipette. Next, the cells were collected in a 1.5
mL Eppendorf tube and incubated for another 5 min at room temperature for more effective cell lysis. 0.2
mL chloroform was added to each mL TRI Reagent and shaken vigorously for 15 s. The mixture was incubated for 15 min at room temperature. The phases were separated by centrifugation at 12,000× *g* for 15 min at 4 ${}^{\circ }$C. The upper aqueous phase containing the RNA was transferred to a new tube. 500 μL 2-propanol was added to the aqueous phase. After vigorously mixing, the samples were incubated for 10 min at room temperature. The precipitated RNA was separated by centrifugation at 12,000× *g* for 10 min at 4 ${}^{\circ }$C. The supernatant was discarded. The RNA pellet was washed with 75% (*v/v*) ethanol and centrifuged at 7500× *g* for 5 min at 4 ${}^{\circ }$C. The supernatant was discarded, and the RNA pellet was air dried for 5–10 min at room temperature. Finally, the RNA pellet was resuspended in 30– 50 μL RNase-free water. RNA amount was assessed by spectrometric analysis on the NanoDrop Lite spectral photometer at 260 nm. Furthermore, protein contamination of RNA was detected by ratio analysis of 260 nm to 280 nm spectrometric analysis. All samples below a ratio of 1.8 were not used for experiments.

### 2.8. RNA Sequencing

Deep RNA sequencing was performed at the European Molecular Biology Laboratory (EMBL, Heidelberg, Germany) with paired-end 2 × 75 nucleotide reads. The removal of rRNA was performed with the Ribo-Zero Gold (Epicentre Biotechnologies, Madison, Wisconsin, USA; Cat# RS-122-2301) or NEBNext rRNA Depletion kit (New England Biolabs, Frankfurt, Germany; Cat# E6310L).

### 2.9. Study Approval

The present study was approved by the ethics committee, Medical Faculty Heidelberg (appl. no. S-390/2011) and participants gave written informed consent. In this study, symptomatic cardiomypathy patients were consecutively, prospectively enrolled. Patients with history of uncontrolled hypertension, myocarditis, regular alcohol consumption, illicit drug use, or cardio-toxic chemotherapy were excluded.

### 2.10. RNA-seq Read-Mapping

Quality clipping and adapter removal of RNA-seq data were performed with Flexbar (version 3.5.0) [32], removing all bases with a Phred score < 28. All reads with potential nuclear-encoded rRNA origin were removed by comparison against the human 45S precursor sequence (NR_046235.1) using Bowtie2 (version 2.3.5.1) [33]. All remaining reads were aligned to the Ensembl *Homo sapiens* genomic reference sequence version 90 using the STAR read-mapping software with support for chimeric alignments (version 2.6.1d) [34] (Appendix B
Figure A3).

### 2.11. CircRNA Detection

Transcript back-splicing junctions with at least 2 supporting reads in at least 2 samples were identified with DCC [35], which is part of the circtools software [36] (version 1.1.0.8), using a RepeatMasker processed version of the Ensembl 90 genome sequence as, described in [37]. For hiPSC-CMs and HUVEC samples, circRNA detection was performed for RNase R-treated and untreated samples. For the human heart samples, only RNase R-treated samples were used (Figure 1B).

### 2.12. Test for Enriched CircRNAs

Enrichment of BSJs in the RNase R-treated samples compared to the untreated samples was performed for the hiPSC-CMs and HUVEC samples by contrasting circular to linear host gene counts using a beta-binomial model implemented in circtools [35,36] and using an adjusted *p*-value cutoff of 0.05 (Figure 1B).

### 2.13. Extraction of RNase R-Resistant CircRNA Exons

Tests for differential exon usage between RNase R-treated samples and untreated samples were performed by circtools [36], employing a negative binomial generalized log-linear model fit at the exon level [38] using an adjusted *p*-value cutoff of 0.05 (Figure 1B).

### 2.14. Identification of circRNAs Enriched for m^6^A-Methylation

RNA-seq data from [27] that captured m^6^A-methylation by m^6^A immunoprecipitation was employed to identify potentially m^6^A-methylated circRNAs. The data from human hearts was processed with the described circRNA detection workflow. Statistical testing for BSJs enriched for m^6^A-methylation compared to control was performed with the described circtools circtest module using an adjusted *p*-value cutoff of 0.05.

## 3. Results

### 3.1. Expressed CircRNAs in Human Cardiovascular Cell Models and Cardiac Tissue

We employed the circtools software [36] to identify circRNA candidates by BSJ reads. The initial detection step yielded 44,063 circRNAs for hiPSC-CMs (Appendix A), 11,124 circRNAs for HUVECs (Appendix A), and 13,327 circRNAs for human hearts (Appendix A) using a cutoff of at least 2 supporting reads per BSJ. The total number of BSJ per sample was comparable between samples of one model system and between the model systems as well as the distribution of reads supporting each BSJ (Figure 2A).

After subsequent filtering for consistent read support and a minimal proportion of junction-spanning reads (1%), we tested for significant enrichment of circRNAs in the RNase R-treated samples compared to the untreated samples. This confined the sets to 9854 enriched circRNAs for hiPSC-CMs (Appendix A) and 2073 enriched circRNAs for HUVECs (FDR <0.05, Appendix A). Overall, we observed a significant (*p*
<0.001) enrichment of backspliced reads in the hiPSC-CM and HUVEC samples (Figure 2B).

For the human heart samples, due to variable RNA integrity, only RNase R-treated samples were processed, therefore no enrichment was calculated. Similar to the hiPSC-CM samples, human samples were aggregated and combined analyses were performed. Alternative splicing was assessed, but was found to only occur at low levels (hiPSC-CMs: 3.4%; HUVECs: 1.5%; human hearts 1.8% with at least >3 supporting reads for alternative splicing variants).

Next, we analyzed the location of the detected circRNAs within their respective host genes. In general, most circRNAs are located within the CDS region, with similar fractions of circRNAs extending in either 5${}^{\prime }$ or 3${}^{\prime }$ UTR and only a minor portion extending from the 5${}^{\prime }$ UTR all the way to the 3${}^{\prime }$ UTR (Figure 2C). Few circRNAs were categorized as not annotated. Of those, around one-third was located on the opposite strand of annotated genes, while two-thirds were of intergenic origin.

We subsequently compared the identified circRNA candidates between all three model systems based on the BSJ coordinates to gain insight into shared subsets of circRNAs. Overall, 1314 circRNAs are jointly expressed in all model systems (Figure 2D). Additionally, we observed a considerable group of shared circRNAs between human hearts and hiPSC-CM (4111 + 1314), which can be expected, given the close relation between these two systems. Interestingly, over 80% of the detected and enriched HUVEC circRNA candidates are also shared with the human hearts and/or the hiPSC-CMs, leaving only 252 distinct HUVEC-specific circRNAs. In contrast, hiPSC-CMs and human hearts both show large numbers of exclusively expressed circRNAs (3874 and 6672, respectively).

### 3.2. Internal CircRNA Structures

After identifying the location of circRNAs within genes, we were interested in how the circRNAs are distributed throughout the set of host genes, i.e. how many circRNAs are originating from each host gene in the different biological systems. We aggregated the number of circRNAs per host gene and additionally factored in diverging sequencing depths throughout the samples. The analyses show a significant (*p*
<0.001) difference between the number of detected circRNAs per host gene between hiPSC-CM, HUVEC, and human heart samples (Figure 3A), even when taking sequencing depth into account (Appendix B
Figure A3).

The definition of circRNAs by back-splicing coordinates on the linear genome is not able to carry all information that defines the sequence of a specific circRNA. In essence, even if a circRNA is detected in two different samples with the same start and end coordinates, no assumptions should be made on the sequence of the circRNA. The RNase R-treated datasets facilitate the identification of RNase R-resistant exons, a hallmark of exon inclusion into circRNAs. We contrasted the number of exons within the BSJ interval from the Ensembl annotation with RNase R-resistant exons and compared the respective exon counts across both categories (Figure 3B). Our analyses show that by adopting the default annotation, the number of exons per circRNA is significantly overestimated when compared to the number of RNase R-resistant exons per circRNA.

By focusing the circRNA analyses on exonuclease-resistant exons, we gained a more fine-grained picture of expressed circRNAs. Using circtools, we identified 185,455 RNase R-resistant exons for hiPSC-CMs (Appendix A) and 98,869 for HUVECs (Appendix A). Briefly, this exon-level approach allows further analyses of circRNAs that are common to HUVECs and hiPSC-CMs and exhibit identical BSJs. Internal alternative splicing of circRNAs has clear consequences for potential functions of a circRNA due to an altered sequence when compared to the annotated exons or circRNAs from different tissues, for an example a circRNA of the SEC62 host gene, that lacks exon 2 in hiPSC-CMs when compared to HUVECs (Figure 3C). By comparing all circRNAs expressed in both hiPSC-CMs and HUVECs on the exon level, we identified 17 circRNAs that show indication of differential splicing of exons (Table 1, Figure 3C). Additionally, the exon-level perspective illustrates that circRNAs from the same host gene may exhibit different BSJs in different conditions. On the one hand, exons that are specifically depleted in hiPSC-CMs can yield to a shifted circRNA start compared to HUVEC, as shown for circSLC33A1 in Figure 3D. On the other hand, circRNAs like TFPI are not expressed at all in hiPSC-CMs, which can be traced back to all of the circRNA’s exons being depleted by RNase R (Figure 3E).

### 3.3. Potential Interaction of Subsets of CircRNAs

To address putative interactions of circRNAs with miRNAs, we overlaid our candidate circRNAs with ≈4000 Ago2 binding sites from hearts of cardiomyopathy patients generated by Spengler et al. [28]. To assess possible interactions regarding RNA translation, we employed ribosome profiling data originating from healthy and cardiomyopathy patient’s hearts [26] in combination with m^6^A methylation data similarly, originating from healthy and cardiomyopathy patient hearts [27]. While m^6^A-methylation has many roles, it may be linked to circRNA translation [45]. We and others previously presented first landscapes of circRNAs expressed in the heart [20,23] and showed that circSLC8A1 is the highest expressed circRNA in the heart. Recent work additionally suggests that circSLC8A1 may act as sponge for mir133a in cardiac myocytes [46]. Indeed, circSLC8A1 is predicted by RNAhybrid [47] to harbor multiple potential binding sites. However, when performing the same analysis with TargetScan [48] only three binding sites were predicted. Compellingly, two of the TargetScan-predicted binding sites, as well as four of the RNAhybrid predictions, overlap with the Ago2 peaks superimposed on our dataset (Figure 4A). In addition to potential interactions with miRNAs, the presence of ribosome protected footprinting reads on the BSJ of circSLC8A1 suggests a direct interaction with the ribosome. Furthermore, although circSLC8A1 was flagged as m^6^A-methylated (Figure 4A), there is no peptide evidence for translation of circSLC8A1 [26]. We expanded the detailed inspection of circSLC8A1 to the complete dataset of 5431 circRNAs that are expressed in hiPSCM-CM as well as in the human heart samples. Of the 39 circRNAs found to be associated with ribosomes, 30 (75%) are also detected in human hearts as well as hiPSC-CMs (Figure 4B, Appendix A). Moreover, we assigned 70% of the m^6^A-methylated circRNAs detected in human hearts from [27] to their counterparts in human heart and hiPSC-CM data of this study (Figure 4B, Appendix A). Ago2 binding in contrast seems to be less common, as we were able to only assign 12% of the 4000 Ago2 peaks to overall 362 circRNAs (Figure 4B, Appendix A).

### 3.4. AUG CircRNAs in hiPSC-CMs and Cardiovascular Cell Models

Stagsted and colleagues recently described a new sub class of circRNAs, termed AUG circRNA [49]. The hallmark of this novel sub class of circRNAs is the inclusion of the start codon of the hosts gene canonical coding sequence as part of the circRNA body, an idea similar to the earlier discussed so-called mRNA trap [50]. Therefore, circRNAs of this sub class by definition contain a start codon that could potentially act as a start of translation for these circRNAs. However, no evidence for peptides derived from translation of AUG circRNA was found in the study.

Here, for the first time, we evaluate AUG circRNAs in hiPSC-CMs and in HUVECs. While we did not observe increased rates of alternative splicing, we recognize a significant trend of higher expression levels of AUG circRNAs compared to non-AUG circRNAs in both hiPSC-CMs and HUVECs (Figure 5A), which is in line with the findings of Stagsted and colleagues [49]. Based on their recent findings, we aimed to explore further roles for AUG circRNAs, given that they contain start codons of their host genes. We employed once again ribosome profiling data from 15 healthy human hearts and 65 DCM hearts [26] to assess the translation of AUG circRNA host genes and non-AUG circRNA host genes in hiPSC-CMs and HUVECs based on P-site counts that represent the amino acid that is linked with the translated polypeptide chain. Intriguingly, we observed reduced RNA translation of AUG circRNA host genes compared to non-AUG host genes in data from healthy hearts as well as data from DCM hearts (Figure 5B). The same trend was observed when comparing translation of HUVEC AUG and non-AUG circRNA hosts genes in the human heart data (Figure 5C). Moreover, we performed analyses of the distance to the next Alu elements, as those have been shown to be linked to circRNA biogenesis [19]. We consistently observed significantly longer distances to the pair of reverse complementary Alu repeats for AUG circRNAs when compared to non-AUG circRNAs for all three model systems (human hearts: *p*
<0.01, HUVECs: *p*
<0.05, hiPSC-CMs: *p*
<0.001, Figure 5D). When focusing on the top 1000 RNase R-enriched circRNAs of hiPSC-CMs regardless of their AUG status, we observed a less pronounced difference compared to AUG circRNAs, demonstrating that the RNase R enrichment here does not seem to correlate strongly with expression level. Directly linked to the Alu distance is the length of the flanking introns of circRNAs. Measuring the intron length revealed significant longer introns flanking AUG circRNAs compared to non-AUG circRNAs (human hearts: *p*
<0.001, HUVECs: *p*
<0.001, hiPSC-CMs: *p*
<0.01, Figure 5E). Next, we analyzed the distance of mRNAs without annotated circRNAs to the next Alu repeat and the length of the flanking introns. Here, we observed significantly longer distances to the next Alu repeat (human hearts: *p*
<0.001, HUVECs: *p*
<0.001, hiPSC-CMs: *p*
<0.01, Figure 5D), as well as significantly longer flanking introns (human hearts: *p*
<0.001, HUVECs: *p*
<0.001, hiPSC-CMs: *p*
<0.01, Figure 5E). Lastly, we asked if host genes of AUG and non-AUG host genes show enrichment for specific functions. Indeed, enrichment analyses with Enrichr [51] show AUG circRNA host genes are enriched for protein modifications such as ubiquitination and polyubiquitination, whereas in contrast, non-AUG circRNA host genes show an enrichment for RNA binding, especially in HUVECs (Table A1, Table A2, Appendix B
Figure A1, Appendix B
Figure A2).

### 3.5. CircRNA Conservation Throughout Multiple Species

Initial analyses suggest that AUG circRNAs are more likely to be conserved than non-AUG circRNAs [49]. We checked the conservation status of circRNAs by lifting the coordinates of human genome (hg38, Ensembl 90) to the pig genome (Sscrofa11, Ensembl 90), thus including a large animal model into our study. Strikingly, circRNAs conserved between human and pig exhibit higher expression than the set of all AUG circRNAs in hiPSC-CMs, with the subset of AUG circRNAs conserved between human and pig showing the highest expression levels (Appendix B
Figure A4). Moreover, we confirmed the AUG circRNAs detected were more likely to be conserved (*p*
<0.01). We extended the conservation analysis to circRNAs shared between human, pig, and a murine model [20] on circRNA and host gene level. From 567 murine heart circRNAs that originate from 440 host genes, 182 circRNA from 154 host genes could be successfully lifted to the human genome (Figure 6A, Appendix A). In order to perform analyses as stringently as possible, only circRNAs that were significantly (*p*
<0.05) enriched in RNase R-treated hiPSC-CMs, pig heart, and murine heart were taken into account. Conservation levels between murine heart and pig heart were comparable to human and murine heart, with 99 circRNAs from 76 host genes (Appendix A). Due to the higher number of enriched circRNAs in human and pig hearts, we expected a higher number of conserved circRNAs between these two species, and indeed, observe 1060 circRNAs from 648 host genes (Appendix A). We performed a more stringent analysis to generate the full picture of circRNAs shared between any of the species. Of the 567 murine circRNAs, 229 (39%) were directly mapped to their counterparts in human and/or pig, while human and pig have 1060 circRNAs in common (Figure 6B). The numbers of distinct circRNAs for human and pig were higher (4241 and 2305 respectively) when compared to mouse (374, Figure 6B). It is notable that the circRNAs that can be computationally mapped to human and pig and that were detected in sequencing data of these species is only a small subset of the number of computationally mapped circRNAs. Overall, we were able to identify 63 circRNAs from 50 host genes that were shared and enriched in RNase R-treated samples throughout mouse, pig, and human (Appendix A, Appendix B
Figure A5), therefore representing a valuable resource and start point for studying circRNAs in the cardiovascular system. In addition, further analyses revealed that of these 63 circRNAs, 34 are part of the AUG sub class.

## 4. Discussion

In this study, based on deep sequencing data, we laid out the circRNA expression landscape of two cardiovascular cell models, hiPSC-CMs and HUVECs, as well as human cardiac tissue.

We identified over 1000 circRNAs identical with regards to their genomic coordinates that are shared between the three cardiovascular model systems in this study. A large overlap between hiPSC-CMs and human heart tissue was to be expected [8], since cardiac myocytes make up approximately 75% of the myocardial tissue volume [52]. However, we found that the majority of HUVEC circRNAs is also shared with human cardiac tissue and hiPSC-CMs. While overlap between HUVECs and heart tissue may be explained by the composition of the heart tissue samples that are not free of endothelial cells and thus may contain circRNAs from endothelial cells and cardiac myocytes, we see similar numbers for hiPSC-CMs and endothelial cells. This could point to a core set of shared circRNAs between these cell types.

Throughout this work, we employed RNase R enrichment of circRNAs prior to sequencing. Our results show that this step, which establishes the exon composition of the circRNA, not only allows increased confidence in detected circRNA candidates, but also may have considerable impact on the functional assessment of these candidates. We present evidence that by only employing known annotations, the numbers of exons, and thus the internal sequence of circRNA candidates may be incorrect. This discrepancy between the sequence predicted from the genome and the sequence of the circRNA molecule may contain essential features, such as binding sites, and will therefore result in distorted results [17].

### 4.1. Assessment of CircRNA Interactions

Previous studies showed that m^6^A-methylation seems to play a role in the translation of a defined subset of circRNAs [45,53]. The comparison of our data with m^6^A-methylated circRNAs [27] suggests, that some circRNAs might have the potential to be translated. Moreover, we observed an overlap of m^6^A-methylated circRNAs with m^6^A-methylated linear transcripts that might influence the stability of the linear transcripts [54]. However, aside from the translation aspect, recent studies also find m^6^A-methylated circRNAs in the context of circRNA immune response, thus implying further roles of m^6^A-methylated circRNAs besides translation [55].

Recent studies found that circRNAs act as template for circRNA-specific peptides [15,16]. The comparison of our data with ribosome profiling data from human hearts [26] suggests that a small number of circRNAs is associated with ribosomes. However, the limited size of the overlap of ribosome-associated circRNAs with circRNAs detected in this study seems to indicate that translation is less prevalent in the circRNA landscape presented here.

The circRNA CDR1as/ciRS-7 has been shown to harbor more than 70 miR-7-binding sites [5,12] and as such paved the way for numerous further studies investigating the potential of circRNAs to sponge miRNAs [13,56]. The comparison of the circRNAs detected in this study with Ago2 CLIP data [28] yielded a small subset of circRNAs that overlap with Ago2 binding site coordinates. The data generated in this study can provide a starting point in that circRNAs of the conserved core set have been checked for Ago2 binding site coordinates, making it possible to quickly check if a circRNA of interest might bind miRNAs through association with the RISC complex. However, further in vitro experiments will be required to confirm these predictions.

### 4.2. AUG circRNAs Exhibit Specific Features

To our knowledge, this is the first study which analyzed the novel class of AUG circRNAs in the cardiovascular context. The members of the AUG class identified here exhibit features that distinguish them from non-AUG circRNAs. Cardiac AUG circRNAs show consistently longer distances to the next Alu element, additionally also significantly longer flanking introns. These observations may have implications on the biogenesis mechanism of this circRNA sub class [49]. Furthermore, AUG circRNAs seem to be enriched for Ago2 interaction and m^6^A-methylation. AUG circRNAs were shown to be expressed at higher levels than their non-AUG counterparts, but in this study we additionally observed potential negative regulatory effects on the level of translation of host genes of AUG class circRNAs, thus shifting the focus from translated circRNAs to the effect of circRNAs on host gene translation. Taken together AUG class circRNAs are distinguishable from non-AUG circRNAs in several key aspects, making them an auspicious target for further studies.

### 4.3. A Highly Conserved Subset of Cardiac CircRNAs

CircRNA conservation is well-studied [5,19]. However, in this study we provide a subset of circRNAs that is not only conserved on the BSJ level, but is also exclusively comprised of circRNAs that have been shown to be enriched upon RNase R treatment. These circRNAs are conserved in human heart tissue, hiPSC-CMs, as well as mouse and pig heart and are promising potential targets for further studies because their function is likely conserved and therefore may be important in the development and progression of CVD. Furthermore, due to their conservation, it is possible to study them in mouse and large animal models as well as in human-derived cardiac cells.

### 4.4. Deciphering CircRNA Sequences

The majority of described circRNAs is currently only defined on back-splice level with very limited knowledge of their sequence, with the exception of single-exon circRNAs. Moreover, knowledge of the exact exon composition of circRNAs is still limited by the length of the circRNA and becomes more challenging with circRNAs that contain numerous or long exons. By employing RNase R enrichment and exon-level predictions, this study aims to extend our knowledge of circRNA sequences. Ultimately however, either very long reads or manual validation in the lab are needed to provide certainty of the exact sequence of more complex circRNAs.

### 4.5. Translational Aspects

The main aim of this study is to provide researchers with specific tools and data for cardiovascular circRNA research to translate their findings into potential therapies or diagnostic tools. The higher stability of circRNAs compared to linear transcripts might allow circRNAs to serve as potential biomarkers that could be used, in addition to, or as novel, markers for different CVDs. Examples include first studies for biomarkers for myocardial infarction [57], atrial fibrillation [58], and coronary artery disease [59]. CircRNAs have also been detected in exosomes [9], thus potentially further extending their potential as biomarkers. However, knowledge of the complete sequence and structure of a circRNA is a prerequisite to design assays that are able to detect the potential circRNA biomarker. In this way, findings of this study may prove useful for the immediate development of circRNA biomarkers.

A growing number of studies implicate specific circRNAs in the development of cardiac hypertrophy [56,60], dilation [61,62], and myocardial infarction [63,64,65]. While the expression of a circRNA might be dynamically regulated in the disease setting, less is known about the functions of circRNAs during disease. A number of studies mechanistically find miRNA sponging as probable cause for both protective and detrimental effects. For example, it was shown that the heart-related circRNA (HRCR) can protect the mouse heart from pathological hypertrophy and heart failure by sponging mir223 [56]. These mechanistic insights are first steps to potential therapeutic use or targeting of circRNAs as treatment strategy. While probably only a minority of circRNAs function as miRNA sponge, linking circRNA data with Ago2 data, as performed in this study, in the long term might help to identify auspicious candidates for future studies.

While recent studies detected only mild changes in the expression of circRNAs in hypertrophic and dilated cardiomyopathy compared to healthy controls [22], often circRNAs of the Titin gene are among those circRNAs that indeed change expression during disease progression. Due to the size of the Titin gene, many circRNAs originating from Titin have rather complex exon structures and thus are harder to study. However, focused studies of circRNAs known to change with disease progression are required to fit circRNAs into the bigger picture of CVD.

### 4.6. Conclusions

Overall, our data provides evidence for differential internal circRNA splicing upon further examination beyond the level of BSJ-conserved circRNAs. We present a strictly filtered set of circRNAs that is precisely conserved and jointly enriched upon RNase R treatment. Moreover, we define the class of AUG circRNAs in the cardiovascular context and contribute further potential regulatory aspects of this new circRNA subclass. Taken together, we believe that the analyses performed in this study contribute deeper insights into potential circRNA functions and arm cardiovascular researchers with new tools to study circRNAs especially in the context of CVD.

## Figures and Tables

**Figure 1 cells-09-01616-f001:**
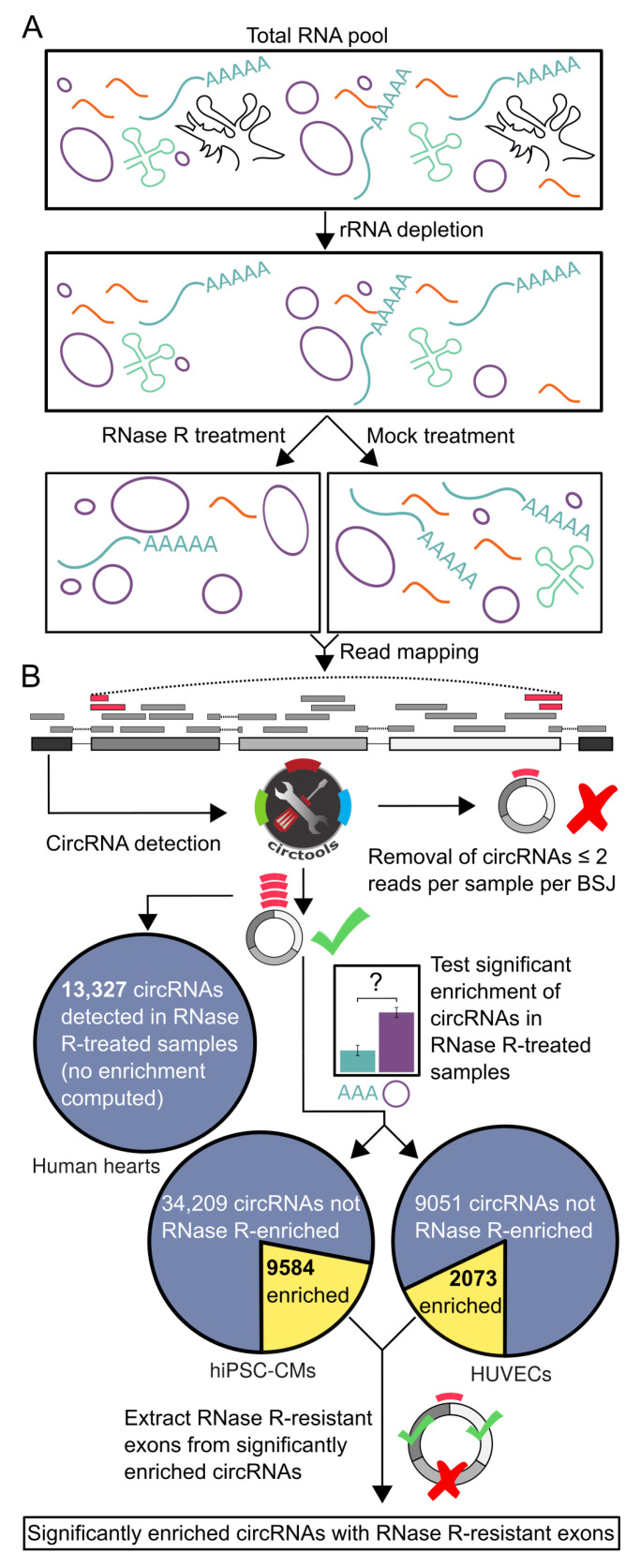
Workflow of library preparation for circRNA sequencing setups and initial steps of the computational pipeline. (**A**) The total RNA pool was depleted of ribosomal RNA (rRNA) and subsequently libraries were constructed from RNase R-treated and untreated samples. (**B**) After read-mapping, circRNAs are detected from chimeric reads (reads). Only circRNA candidates with ≥2 reads per BSJ are either directly employed for analysis (human heart dataset) or tested for significant enrichment (hiPSC-CMs and HUVECs).

**Figure 2 cells-09-01616-f002:**
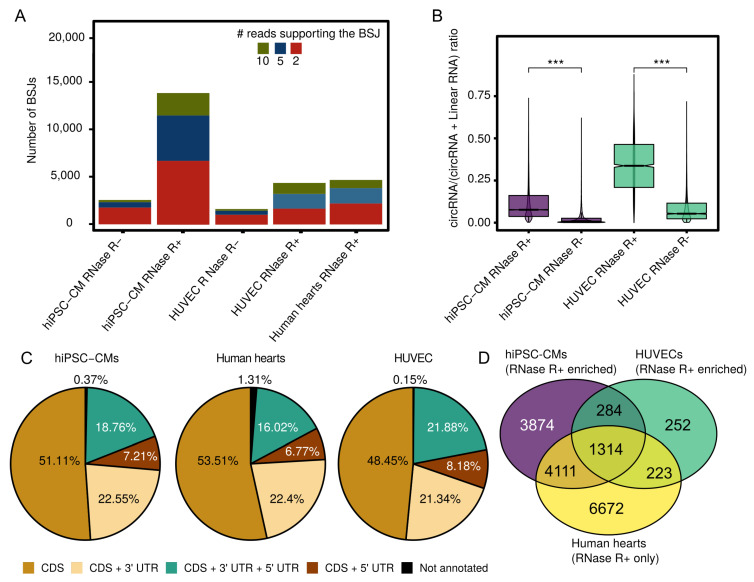
CircRNA detection, RNase R enrichment, location, and overlap. (**A**) absolute numbers of detected BSJs per sample. Dark green: ≥10 reads supporting the BSJ, dark blue ≥5 supporting reads, red ≥2 supporting reads. (**B**) Assessment of the RNase R treatment for hiPSC-CMs and HUVECs via circRNA/(circRNA + linear RNA) ratios. (**C**) Distribution of the origin of circRNAs within gene bodies and not annotated regions. (**D**) Shared and distinct subsets of circRNAs based on the BSJ coordinates. Wilcoxon rank-sum test: ***, *p*
<0.001.

**Figure 3 cells-09-01616-f003:**
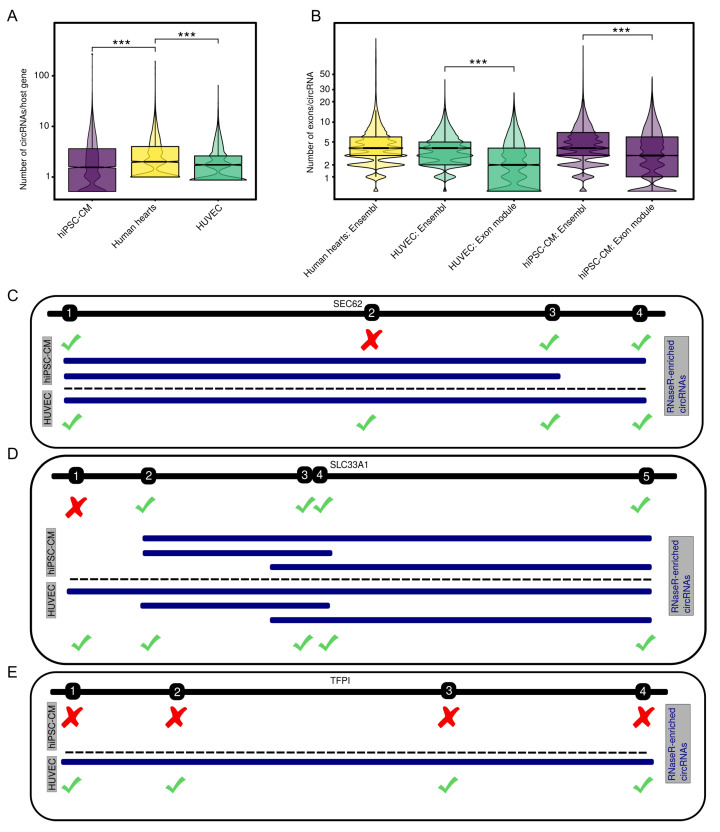
Number of circRNAs per host gene, exon composition, and differences of circRNA sequences across samples. (**A**) Detected circRNAs per host gene normalized to sequencing depth. (**B**) Divergence between predicted number of exons per circRNA based on the Ensembl 90 annotation and the number of RNase R-resistant exons. (**C**) circRNAs of SEC62 (blue) are depleted for exon #2 in hiPSC-CMs when compared to HUVECs. (**D**) circRNAs of SLC33A1 (blue) show different backspliced exons in hiPSC-CMs and HUVEC cells, yielding a different circRNA starts. (**E**) A circRNA of TFPI (blue) is completely depleted in hiPSC-CMs when compared to HUVECs. Green check mark: exon RNase R-resistant, red cross: exon not RNase R-resistant, black: exons of the host gene, numbering refers to position in circRNA. Wilcoxon rank-sum test: ***, *p*
<0.001.

**Figure 4 cells-09-01616-f004:**
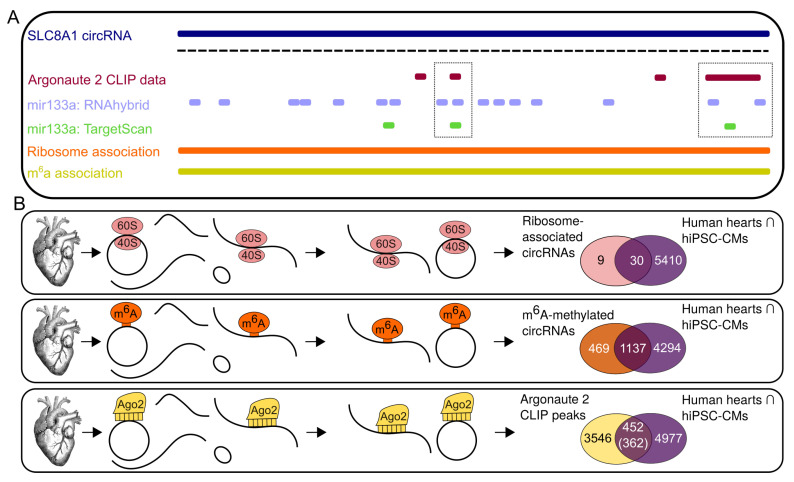
Combination of multiple datasets as tracks and generation of track data. (**A**) Graphical representation of the circSLC8A1 locus, showing Argonaute 2 (Ago2) peaks, mir-133a binding site prediction by RNAhybrid and TargetScan (dotted boxes), the predicted single-exon circRNA, association with ribosomes, and m^6^A-methylation status. (**B**) Workflows of sample generation for ribosome-associated circRNAs, m^6^A-methylated circRNAs, and circRNAs associated with Ago2. Venn diagrams show overlap of the circRNAs with the joint circRNAs from hiPSC-CMs and human hearts. For the Ago2 overlap, the upper number refers to the number of peaks, the number below refers to the number of circRNAs.

**Figure 5 cells-09-01616-f005:**
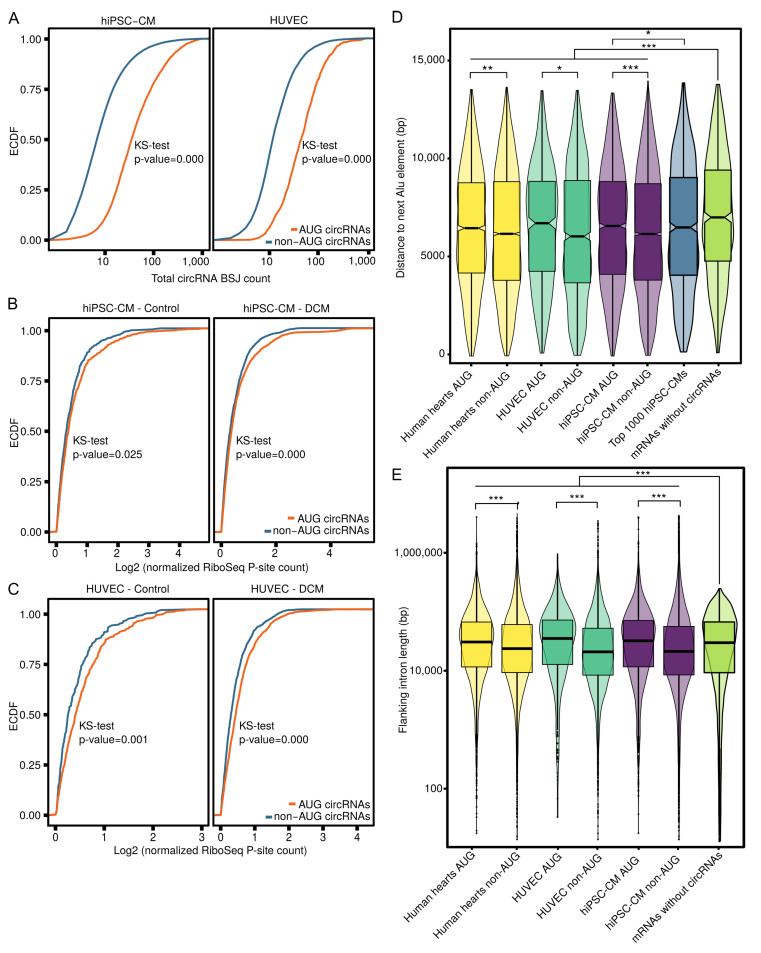
Properties of AUG circRNAs and host genes of AUG circRNAs. (**A**) Expression of circRNAs of the AUG class (orange) and non-AUG class (blue) in hiPSC-CMs and HUVECs shown as empirical cumulative distribution function (ECDF). (**B**) Assessment of translation of host genes of AUG circRNAs (orange) and host genes of non-AUG circRNAs (blue) in hiPCM-CMs in 15 healthy human hearts and 65 DCM hearts. (**C**) Assessment of translation of host genes of AUG circRNA (orange) and host genes of non-AUG circRNAs (blue) in HUVECs in 15 healthy human hearts and 65 DCM hearts. (**D**) Distance to next Alu repeat from circRNA blacksplice junction for AUG and non-AUG circRNAs. (**E**) Length of adjacent introns for AUG and non-AUG circRNAs. KS test (Kolmogorow–Smirnow test): *, *p*
<0.05; **, *p*<0.01; ***, *p*<0.001.

**Figure 6 cells-09-01616-f006:**
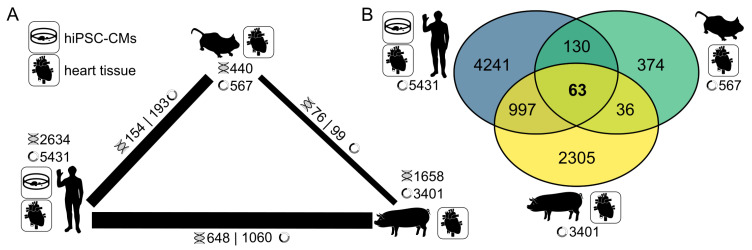
Conservation of circRNAs in three model species of cardiovascular research. (**A**) circRNAs conserved on back-splicing level and additionally enriched in RNase R-treated samples between mouse heart, human heart, hiPSC-CMs, and pig heart. (**B**) Sizes of the shared subsets and exclusive circRNAs of the strongly (positionally) conserved circRNAs shown in (**A**) for each of the assessed species.

**Table 1 cells-09-01616-t001:** List of host genes giving rise to circRNAs that show different internal exon composition when comparing HUVEC and hiPSC-CMs. Gene names with * additionally indicate that the exon-containing circRNA is known to be m^6^A-methylated.

Gene	Gene Description	Note
PRRC2C	Proline Rich Coiled-Coil 2C	-
EPS15 *	Epidermal Growth Factor Receptor Pathway Substrate 15	Highly expressed in heart
TBC1D8	TBC1 Domain Family Member 8	-
NDUFS1	NADH-ubiquinone oxidoreductase 75 kDa subunit, mitochondrial	Highest expression in heart
LRPPRC	Leucine Rich Pentatricopeptide Repeat Containing	Multifunctional Protein Involved in Energy Metabolism and Human Disease [39]
ESYT2 *	Extended Synaptotagmin 2	Tethers the endoplasmic reticulum to the cell membrane and promotes the formation of appositions between the endoplasmic reticulum and the cell membrane
SEC62 *	SEC62 Homolog, Preprotein Translocation Factor	Mediates post-translational transport of precursor polypeptides across endoplasmic reticulum
NOP14-AS1	NOP14 Antisense RNA 1	LncRNA
G3BP2	G3BP Stress Granule Assembly Factor 2	Involved in isoprenaline-induced cardiac hypertrophy [40]
CUL5 *	Cullin 5	Protein is expressed at its highest levels in heart and skeletal tissue [41]
STRN3	Striatin 3	Regulation through Quaking [42]
USP7	Ubiquitin Specific Peptidase 7	-
TCEA1	Transcription Elongation Factor A1	Promotes activity of the myogenic regulatory factors in skeletal muscle [43]
ATP5F1C *	ATP Synthase F1 Subunit Gamma	Highly expressed in heart
RNF10	Ring Finger Protein 10	Highly expressed in heart
WSB1	WD Repeat and SOCS Box Containing 1	Linked to hypoxia response [44]
SPAG9	Sperm Associated Antigen 9	-

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
