# Peer review of "Deep Characterization of Circular RNAs from Human Cardiovascular Cell Models and Cardiac Tissue"

_cells, 2020, doi:10.3390/cells9071616_

Round 1
Reviewer 1 Report
Jacobi et al; Deep characterization of circular RNAs form human cardiovascular cell models and cardiac tissue.
An extensive characterization of circular RNAs in cardiac cell models. The authors investigate hi-PSC-derived cardiomyocytes, primary cardiac tissue and human umbilical vein endothelial cells (HUVECs). The strengths of this manuscript lies in the thorough characterization of transcript reads likely to be circular RNAs. The weakness of the manuscript is in the descriptive nature of the study; although the authors make possible lists to use to prioritize which circular RNAs are most likely to have a biological function, there are no wet lab experiments to demonstrate effects.
Comments:
The manuscript is well written in, generally, a clear and comprehensible language.
Figures:
Figures can be improved.
Figure 2:
A.Two back-splice junctions (BSJ) is set as the cut off for accepting a circular RNA as existing (line 216-221. It is not clear how the numbers of circular RNA species (BSJs) given in the text relates to Figure 2A.
B. The color scheme of Fig 2B is confusing, as one would automatically tend to think that the colors in Fig 2B corresponds to the same legends as in Fig. 2A, but this is not the case.
Figure 3:
C to E: This sub-figure is very difficult to understand. The fonts and lines are generally too small to be legible unless hugely magnified. It is not clear from the figure, the legend or the text, what is really shown in this figure part. It looks completely like a poor screenshot from a genome browser. Should be redrawn in a more simplistic way OR alternatively then all tracks shown should be explained in the legend. Similarly, for the supplemental figure containing browser screen shots.
Lines 272-9 – the figure reference given is to figure 2C-2E, but it should be Fig 3 C-3E.
Figure 4:
- ‘Argonaut’ should be ‘Argonaute’ (2 places). It is not explained what the two different numbers 452*/362 denote.
Figure 5:
In this figure, the authors investigate properties of cirs that contain the start codon. Please explain in legend what ECDF is short for (Empirical cumulative distribution function?). In A-C the ECDFs for AUG-containing Cirs and non-AUG containing cirs are drawn, showing in A that AUG-containing cirs have a higher expression level. In B and C, the genes having cirs containing an AUG codon are less frequently associated with ribosome footprints indicating that these genes (their transcripts) are translated at lower levels. However, the legend B and C seems reversed.
In D and E, distance to nearby Alu elements and flanking intron length are computed for the groups of cirs having an AUG or not, showing some differences. These two figure panels are uninformative, as asterisks are the only thing indicating differences between the violin bars – it is not possible to visually discern any difference – so figure should be redrawn with different y-scale. It would be nice to have distances computed for mRNAs (first translated exon) not having a circular RNA annotated to it - just being close to the translation start site could perhaps explain the difference observed in C and E.
Minor comments:
Line 121: Something is missing – the sentence does not make sense.
Line 156: Should be DNase
Several places: ‘RNase R-treated libraries’ or similar. Strictly speaking, this makes no sense. The libraries are not RNase treated. That would not change much, as they consist of DNA. Please use order wordings, such as libraries from RNase R-treated samples or similar.
Reviewer 2 Report
This work presents a comprehensive study of Circular RNAs (circRNAs) in cardiovascular cell models and cardiac tissues related to ischemic and dilated cardiomyopathies. The study provides a substantial improvement in our understanding of the role of circRNAs in the development and progression of cardiovascular diseases (CVDs).
The manuscript is well-organized and well-written, the methodology and protocols are clearly described, and the results offer novel insights into circRNA expression landscape in the context of ischemic and dilated cardiomyopathies. I do not have any major comments apart from a few suggestions given below with regard to Discussion:
- The presentation is quite in-depth on the biological side of the topic, but it perhaps can benefit from some discussion on the translational aspects. I suggest devoting a paragraph in Discussion addressing the implications of the findings in this paper for immediate and long-term therapeutics for CVDs.
- Somewhat related to the above comment, and especially from the perspective of readers outside of the cell biology community, can further insights into the role of circRNA improve our understanding of the bigger picture of disease progression? In particular, how one should relate the subcellular events (such as circRNA expressions) to the remodeling at the cellular, tissue, and organ levels in CVDs? Ultimately, the cardiac function is what needs to be fixed, and some thoughts on correlations between function deterioration and circRNA activities will make the paper more appealing for the broader audience in the cardiovascular community.
- I suggest organizing Discussion into subsections with relevant headers like in Results for smoother navigation of Discussion.
